# Metagenomic Profiling of Microbial Pathogens in the Little Bighorn River, Montana

**DOI:** 10.3390/ijerph16071097

**Published:** 2019-03-27

**Authors:** Steve Hamner, Bonnie L. Brown, Nur A. Hasan, Michael J. Franklin, John Doyle, Margaret J. Eggers, Rita R. Colwell, Timothy E. Ford

**Affiliations:** 1Department of Environmental Health Sciences, School of Public Health & Health Sciences, University of Massachusetts Amherst, Amherst, MA 01003, USA; 2Department of Microbiology, Montana State University, Bozeman, MT 59717, USA; franklin@montana.edu (M.J.F.); mari.eggers@montana.edu (M.J.E.); 3Department of Biological Sciences, University of New Hampshire, Durham, NH 03824, USA; bonnie.brown@unh.edu; 4CosmosID Inc., 1600 East Gude Drive, Rockville, MD 20850, USA; nur.hasan@cosmosid.com (N.A.H.); rcolwell@umiacs.umd.edu (R.R.C.); 5Center for Bioinformatics and Computational Biology, University of Maryland, College Park, MD 20742, USA; 6Center for Biofilm Engineering, Montana State University, Bozeman, MT 59717, USA; 7Crow Water Quality Project, Crow Agency, Little Big Horn College, MT 59022, USA; doylej@lbhc.edu; 8Crow Environmental Health Steering Committee, Crow Agency, Little Big Horn College, MT 59022, USA

**Keywords:** metagenomics, pathogen detection, waterborne disease

## Abstract

The Little Bighorn River is the primary source of water for water treatment plants serving the local Crow Agency population, and has special significance in the spiritual and ceremonial life of the Crow tribe. Unfortunately, the watershed suffers from impaired water quality, with high counts of fecal coliform bacteria routinely measured during run-off events. A metagenomic analysis was carried out to identify potential pathogens in the river water. The Oxford Nanopore MinION platform was used to sequence DNA in near real time to identify both uncultured and a coliform-enriched culture of microbes collected from a popular summer swimming area of the Little Bighorn River. Sequences were analyzed using CosmosID bioinformatics and, in agreement with previous studies, enterohemorrhagic and enteropathogenic *Escherichia coli* and other *E. coli* pathotypes were identified. Noteworthy was detection and identification of enteroaggregative *E. coli* O104:H4 and *Vibrio cholerae* serotype O1 El Tor, however, cholera toxin genes were not identified. Other pathogenic microbes, as well as virulence genes and antimicrobial resistance markers, were also identified and characterized by metagenomic analyses. It is concluded that metagenomics provides a useful and potentially routine tool for identifying in an in-depth manner microbial contamination of waterways and, thereby, protecting public health.

## 1. Introduction

Water is essential for human life and productivity, yet both water quality and security are increasingly under threat globally [1,2,3]. Unfortunately, while wealthy countries are able to afford effective water treatment, poorer nations are severely hampered by a lack of resources for safe water to protect public health. In both cases, sources of contaminants entering waterways are not sufficiently addressed [2].

Communities within the Crow Nation in south central Montana have been aware of deteriorating water quality of the Little Bighorn River for many years [4]. Concerned members of the Crow Nation founded the Crow Environmental Health Steering Committee, the mission of which is to research and mitigate issues concerning environmental public health, improve community health and raise awareness among the tribal population. The Steering Committee and tribal elders have also voiced concern about the effects of climate change on water quality and public health [5,6,7]. Concern about a link between climate change and public health is warranted given that studies have indicated that warmer temperatures may promote an expanded range of distribution of vector-borne infectious diseases and extended seasons of transmission [8,9,10].

The Steering Committee has worked closely with faculty and students at the local Little Big Horn College (LBHC) and Montana State University (MSU), who have conducted water quality studies in accord with priorities established by the Crow tribe. A major focus has been to identify and characterize bacterial, chemical, heavy metal, and radionuclide contaminants in domestic well, recreational, and source waters [11,12,13,14,15]. Findings have informed the local public and environmental health efforts, and supported funding procurement to improve sewage and water treatment facilities along a portion of the Little Bighorn River that serves as source water for the community [12,16].

Among the various concerns addressed, Hamner et al. [17] conducted a source tracking study of run-off from a large concentrated animal feed operation (CAFO) located in the headwaters area of the Little Bighorn River. Results of the study demonstrated the presence of several *Escherichia coli* human disease-associated serotypes immediately downstream of the CAFO drainage that matched serotypes in cattle manure from the CAFO. As part of the study, enterohemorrhagic and enteropathogenic *E. coli* (EHEC and EPEC) serotypes were also identified in a popular swim hole of the Little Bighorn River, a summer-time recreational site for children in the town of Crow Agency, the administrative center of the Crow tribe. Fecal contamination and run-off from the CAFO and dozens of smaller ranching operations, as well as leakage from septic systems from homes bordering the river banks, contribute to pollution of the Little Bighorn River and many smaller tributary streams within the watershed.

Recent developments in whole genome sequencing (WGS) technology make it feasible to use DNA sequencing for disease diagnostics and public health surveillance of pathogens [18,19]. Development of portable sequencing platforms, such as the Oxford Nanopore Technologies MinION device, allow for rapid sequencing of whole genomes [20], which facilitates metagenomic sequencing to be accomplished in remote locations [21]. Portability can be especially useful during disease outbreaks where laboratory resources are limited, and when coupled with real-time analysis, can facilitate prompt epidemiological study and public health response to epidemic outbreaks [19,22].

Metagenomic sequencing has been used to study the epidemiology of a variety of infectious disease agents. In one study, a metagenomic approach proved more accurate than conventional genotyping in analyzing an outbreak of tuberculosis, namely by improving identification of single nucleotide polymorphisms and assignment of genome clusters (factors related to the evolution of the outbreak strain) and tracing the spread of the outbreak [23].

Metagenomics currently is used to describe microbial populations in water and sediment to understand community structure and the role of microorganisms in ecological processes [18,24,25,26]. Metagenomics has also been used to examine water quality to protect public health [27,28]. Traditional methods for monitoring water quality focus on fecal coliform counts, but methods employing metagenomics provide additional functional and genomic information for species and strains of microbial pathogens. In addition, markers of the potential for antibiotic resistance, and the presence of virulence genes can also be identified in recreational and source waters [27,28].

Rather than target identification of a pre-selected group of pathogenic microbes or virulence genes by traditional culture, microscopy, immunoassay, or PCR-based methods [29], metagenomics employing next generation sequencing allows accurate identification and characterization of all microorganisms within samples for which genomic data are archived. Further, DNA sequence-based identification and characterization can now be done for microorganisms not easily cultured in a diagnostic setting, and can be used to identify multiple pathogens present in a poly-microbial infection or in water bodies.

## 2. Materials and Methods

Water samples were collected from the Crow Fair swim hole of the Little Bighorn River, Crow Agency, Montana, on July 16, 2017. The swim hole is located at latitude/longitude of 45^o^36′1”N, 107^o^27′12”W, and is a popular summer recreational site used by children and adults of the population of ca. 1,600 residents of Crow Agency. During sampling, several children were observed swimming 100 meters upstream of the sampling site. Four samples were collected at ten-minute intervals over the course of 30 minutes, then pooled. Samples were transported on ice to Montana State University for processing by two different methods. First, 100 mL aliquots from each of the four consecutive samplings were pooled (400 mL total) and filtered using 47 mm, 0.45 μm filters to collect particulates. Filters were processed using the PowerWater DNA isolation kit (Qiagen). (Technical notes: The PowerWater kit was chosen in part due to its incorporation of reagents to remove inhibitors that may interfere with downstream PCR and DNA sequencing reactions. Choice of 0.45 μm filters as opposed to 0.22 μm filters was due to the turbidity of the water samples. We followed the manufacturer’s recommendation to use 0.45 μm filters for turbid water samples to reduce clogging and allow a greater volume of water to be filtered than would be possible with the more restrictive 0.22 μm filters. We readily acknowledge that this choice may have reduced the variety of bacteria detected, since during the initial stages of filtering, smaller bacteria would be lost in the larger 0.45 μm pores, but also are aware that as the filters clogged, many smaller cells should have been captured.) Modifications to the PowerWater kit protocol were made as follows: Filters were placed in a 5 mL PowerWater kit tube, and 1.5 mL of PowerWater kit buffer (instead of 1 mL per manufacturer’s instruction) and Metapolyzyme (20 μL of a 10.0 mg mL^−1^ sterile PBS pH7.5, Sigma #MAC4L) were included in the lysis step to enhance digestion of extracellular material and release of DNA. (Note: In our experience, use of 1.5 mL of the first PowerWater kit buffer was found to increase the yield of DNA compared to using only 1 mL.) The tube was vortexed for ca. 60 s (minimizing fragmentation) and incubated overnight at 37 °C, with periodic rotation and agitation. After overnight incubation, DNA extraction was continued, following PowerWater kit manufacturer’s instructions.

A separate DNA extraction procedure was carried out to harvest DNA for the detection of coliform bacteria. To begin, three technical replicates of ca. 50 mL river water were filtered under vacuum and the filters placed on m-Coliblue24 plates [30] and incubated overnight at 37 °C for coliform counts. Following the manufacturer’s protocol, membrane filters of 0.45 μm pore diameter were used [30] for the m-ColiBlue24 assay. It is noteworthy that consistency was maintained for both filtration procedures, in that 0.45 μm pore filters were also used for filtering the larger 400 mL volume of river water for the uncultured, non-selective procedure described above. After the resulting blue *E. coli* colonies on the m-Coliblue24 plates were enumerated, one filter yielding colony growth was selected for DNA extraction, using the PowerWater kit and the amended protocol described above. A second filter with colony growth was lifted with forceps, replica plated on CHROMagarO157 agar (CHROMagar), and incubated overnight at 37 °C. The appearance of mauve-colored colonies indicated putative EHEC.

DNA sequencing was accomplished using the MinION sequencing platform (Oxford Nanopore) and the 1D Ligation Sequencing Kit (Nanopore kit SQK-LSK108), following the manufacturer’s instructions. Resulting fast5 data files were basecalled using Albacore Version 1.2.6 after which fastq sequence processing and analyses were performed using CosmosID (www.cosmosid.com) and MG-RAST [31] software. The metagenomic data is available in MG-RAST (mg-rast.org, MG-RAST ID numbers mgm4778816.3 and mgm4778817.3).

Because of the heightened degree of public health concern regarding *E. coli* serotypes O157:H7 and O104:H4, and *V. cholerae* O1 El tor, coverage plots and completion estimates were also generated as an additional indication of confidence in identifying these potential pathogens. GraphMap [32] was used for mapping and coverage calculations for these sequences.

DNA preparations from both river water (without selective growth) and m-ColiBlue24 selection samples were examined by PCR for the presence of *eae* and *Stx* genes that are indicative of EHEC and EPEC as previously described [17,33]. Presence of *eae* is characteristic of both EHEC and EPEC; on the other hand, EHEC contains *Stx* genes while EPEC lacks *Stx* genes.

## 3. Results

### 3.1. Metagenomic Sequences Generated from DNA Prepared Directly from River Water without Selective Growth on m-ColiBLue243

Environmental DNA from microorganisms collected by filtration of water from the Little Big Horn River was subjected to shotgun sequencing using the Oxford Nanopore MinION platform. Sequencing generated 397,884 reads comprising ~1.1 Gbp with an average read length of 2760 bp. CosmosID analysis of the DNA sequences indicated the presence of both Eukarya and Bacteria in the river water community. These included: Eukaryotic protists (Table 1), fungi (Table 2), and bacteria (Table 3 and Figure 1). Several eukaryotic genera that are of potential concern to human health, including *Acanthamoeba, Leishmania, Candida, and Rhizomucor*, were identified in the analyses (Table 1 and Table 2). Bacteria of concern to human health, *Acidovorax* and *Aeromonas salmonicida*, were also identified (Table 3). *Limnohabitans* was the dominant genera in the filtered river biomass, followed by *Actinobacterium*, a genus that includes important members of a healthy gut microbiome.

### 3.2. Metagenomic Analysis of DNA Prepared from Filter after Selective Growth on m-ColiBlue24 Medium

The average concentration of *E. coli* was 66 colony forming units (CFU) per 100 mL water that was detected on filters incubated overnight on m-ColiBlue24 medium. This concentration is well below the limit of 126 CFU per 100 mL established by the EPA [43] for recreational water to be considered safe for swimming.

DNA was prepared and sequenced from colonies grown overnight on the filters with an m-ColiBlue24 selection. A total of ~1.6 Gbp of data was generated, comprised of 1,261,165 sequence reads with an average length of 1260 bp. Several bacterial species, some of which are important human pathogens, were detected that had not been identified in the native river water metagenome (Table 4 and Figure 2). In addition to numerous opportunistic pathogens, strains of toxigenic *E. coli*, *Shigella* spp., and *Vibrio cholerae* were also detected.

A number of bacteriophages (Table 5), antimicrobial resistance (AMR) gene markers (Table 6), and virulence genes (Table 7) were identified from the metagenomic analysis after m-ColiBlue24 selection. Several of these bacteriophages, AMR markers, and virulence genes are relevant to human health.

The *eae* and *Stx* genes were both undetected in DNA prepared from unenriched river water, whereas these genes were both detected in DNA isolated from a filter cultured on selective m-ColiBlue24 medium (data from PCR not shown). The presence of *Stx2* converting phage sequences was also indicated by the metagenomics analysis (Table 5). Colonies grown on m-ColiBlue24 media and replica plated onto CHROMagarO157 media gave rise to scattered, small spots of mauve growth, indicating the presence of EHEC bacteria.

Five markers of antimicrobial resistance (AMR) at 18 different gene loci were identified in the metagenomic analysis of the m-ColiBlue24 selection sample (Table 6). These markers are related to efflux of antibiotics, resistance to the beta-lactam class of antibiotics, as well as resistance to ampicillin, fluoroquinolones, and polymyxins.

Several virulence genes that contribute to the ability of microbes to cause disease were identified (Table 7). These genes code for virulence factors related to attachment, acid resistance, enhanced serum survival, the competitive advantage against other microbes, and iron acquisition capability.

Calculations of DNA sequencing coverage and depth of coverage were made by mapping reads to the genomes of three pathogens of major public health significance. Reads were mapped to *E. coli* O104:H4, *E. coli* O157:H7 Sakai, and *V. cholerae* O1 El Tor with 96%, 95% and 93% coverage (completion) of genomes, respectively (Table 8 and Figure 3, Figure 4 and Figure 5). The depth of coverage was 52×, 50×, and 36×, respectively, for the three genomes (Table 8). Based on genome reporting standards proposed by Bowers et al. [69], these genomic coverages would meet the criterion for high quality metagenome-assembled genomes for these three species. Given that the sequences were mapped to reference genomes with high fidelity, there are unlikely to be multiple, heterogeneous populations for each species. Consequently, these pathogenic populations were present in the river water, and were detectable after selection and enrichment on m-ColiBlue24 media.

## 4. Discussion

This study describes a metagenomic analysis of water samples collected from a popular swimming site along the Little Big Horn River during the summer of 2017. This work was predicated on previous detection and identification of EHEC and EPEC bacteria in water samples collected from the Little Bighorn River [17] and ongoing concerns of the local community related to water quality and safety. Initial metagenomic analysis of total DNA isolated from filtered river water indicated the presence of species and strains of typical freshwater microorganisms, including both culturable and non-culturable microorganisms. Distinguishing between culturable and non-culturable microbial strains is important, since a study of freshwater lake bacteria estimated approximately only 0.25% of the total bacterial population was culturable [70]. Indeed, most bacterial populations in these environments are viable but not culturable (VBNC) using standard bacteriological culture methods [71,72,73]. A variety of both naturally occurring and potentially pathogenic bacterial species have been shown to enter the VBNC state in response to environmental stress, reducing detection of a significant percentage of a population with relevance to public health in environmental surveillance.

A second metagenomic analysis was also performed using DNA prepared from a filtered water sample after incubation on m-ColiBlue24 medium overnight to allow for selection of coliforms and related species. This two-pronged approach was taken to enhance detection and identification of pathogens, the growth of which may be inhibited by other river bacteria, and therefore not previously recognized in earlier studies targeting detection of coliforms in the river water [17].

Metagenomic analysis of DNA extracted from filters without growth on selective medium revealed a rich diversity of microorganisms, the predominant species of which are presented in Table 1, Table 2 and Table 3. The absence of *E. coli* in DNA prepared without enrichment on a selective medium was not surprising given the relatively small number of reads and because enrichment on selective media yielded only 66 CFU/100 mL of *E. coli* in overnight culture on m-ColiBlue24 medium. This medium has been approved by the EPA [30,74] as a sensitive method for detecting and monitoring fecal coliform (*E. coli*) bacteria in fresh water, where a count of 126 CFU/100 mL (calculated as geometric mean for samples collected over a 30-day period) for *E. coli* is the maximum permissible limit for recreational waters [43]. Lack of detection of *E. coli* by metagenomic analysis without selective growth is attributed to the overwhelming abundance and diversity of non-*E. coli* microorganisms that were present. The proportion of *E. coli* present in the water samples was representatively small in comparison to the high microbial load on selective media, evidenced by results of the analysis of the m-ColiBlue24-derived metagenome, revealing Gammaproteobacteria and coliform bacteria in significant abundance. Our choice of 0.45 μm pore diameter membrane filters was based on following the manufacture’s protocol [30] for the EPA-approved m-ColiBlue24 method, as well as water sample turbidity. We acknowledge that use of this pore size instead of a smaller pore diameter filter could have resulted in our missing smaller sized microorganisms of public health significance. However, species and strains (Table 4) that were identified, including many serotypes of diarrheagenic bacteria, such as EHEC O157:H7, were also identified in an earlier study [17].

DNA sequences indicative of *E. coli* serotype O104:H4 and *V. cholerae* O1 El Tor, both human pathogens of significant interest, were identified (Table 4). Of particular concern, *E. coli* O104:H4 is an emerging pathogen that first received widespread attention in 2011 as the causative agent of the largest outbreak of Shiga toxin-related disease [75] recorded to date [50]. In Germany and surrounding areas, an O104:H4 outbreak strain caused 3,842 cases of illness, including 18 deaths. Among those stricken, 855 people developed hemolytic uremic syndrome (HUS), leading to an additional 35 deaths [50]. The disease-associated O104:H4 outbreak strain is a novel variant of enteroaggregative *E. coli* (EAEC) that acquired the Shiga toxin gene that is characteristic of EHEC.

Detection of *V. cholerae* sequences in the Little Bighorn River is not surprising. *V. cholerae*, the causative agent of cholera, is an aquatic bacterium with world-wide distribution [76], that may be due to globalization and may indicate changing human demographics. Recently, *V. cholerae* caused an outbreak of disease in Haiti that had not been seen in 100 years [77,78]. Several virulence genes have been reported as essential for these bacteria to cause an outbreak of cholera, especially including the *ctxA* and *ctxB* genes encoding cholera toxin and carried by the bacteriophage CTXφ. This bacteriophage was not detected in this study (Table 5). However, a cluster of genes (VCA0107, VCA0109, VCA0111, VCA0121, *vgr*G-3, and *vas*H; see Table 7) associated with the type VI secretion system (T6SS), an important virulence factor of many Gram-negative pathogenic bacteria, including *V. cholerae* [79], were detected. In the related species *V. proteolyticus*, the T6SS includes cytotoxic effectors that target both prokaryotic and eukaryotic cells [80]. In *V. cholerae*, the T6SS has been shown to kill other bacterial species, releasing DNA that in turn can be taken up in the process of horizontal gene transfer (HGT) by naturally competent *Vibrio* bacteria [67]. Genes taken up by HGT may enhance the antibiotic resistance and virulence potential of *Vibrio* cells, highlighting the evolutionary potential of pathogenic bacteria in natural environments to become more virulent.

Of relevance to human health, bacteriophage-encoded genes that enhance the pathogenicity of host bacteria were also detected. Two types of Shigella-specific phage, SfII and SfIV, allow for O antigen modification and increased antigen variation [59,60]. The *Stx2* converting phage of *E. coli* O157:H7 and other related Shiga toxigenic *E. coli* (STEC) encodes the Stx2 protein, an important virulence factor causing lysis of host cells and contributing to hemolytic uremic syndrome [61].

Detection of several AMR markers in the m-ColiBlue24 metagenome (Table 6) is relevant as the worldwide spread of antibiotic resistance is increasingly recognized as a major public health threat, compromising treatment of a variety of infectious diseases [81]. Widespread use of antibiotics in human and veterinary medicine has contributed to an increasing pool of bacteria harboring AMR genes and these bacteria, in turn, are now widely distributed in agricultural products, animals, humans, and the environment [82].

The metagenomic analyses presented in this study indicate that a variety of potential human disease-related pathogens and AMR markers were present and detectable in water samples collected from the Little Bighorn River during the summer of 2017. The presence of gene markers for *E. coli* O157:H7 (Table 4 and Table 5), a human pathogen of significant concern, is in agreement with earlier findings of Hamner [17]. Presence of Shiga toxin gene markers indicated by both PCR (data not shown) and metagenomic analysis, as well as mauve-colored colony growth on ChromagarO157 medium, a differential/selective medium and indicative test for O157:H7, provide both genetic and phenotypic evidence for continued presence of *E. coli* O157:H7 bacteria in the Little Bighorn River. As it is understood that the major reservoir of O157:H7 bacteria is cattle and other ruminants [83], livestock ranching operations along the length of the Little Bighorn River, including a large concentrated animal feed operation close to the headwaters of the river, provide likely sources of this contamination to the watershed.

Penicillin derivatives are widely used in animal husbandry and hence ampicillin and beta-lactamase resistance might be expected to coincide with the presence of animal-associated pathogens [84]. However, tetracyclines tend to be more broadly used, and the absence of any tetracycline resistance gene markers would suggest further work is needed to identify sources of contamination. It is not currently known which antibiotics are primarily used in the Little Bighorn watershed.

Animal experiments with the *E. coli* O157:H7 bacteria or other potential pathogens identified in the present study were not conducted. Therefore, it is unclear whether isolates from the Little Bighorn River are capable of causing disease. Nevertheless, the presence of *E. coli* O157:H7 bacteria detected in the river consistently and over several years, along with identification of other known pathogens, is of concern. Consequently, the potential for horizontal gene transfer based on detection of AMR genes and evolution of pathogens with enhanced pathogenic potential and spread of AMR cannot be ignored [85].

The metagenomics analyses carried out in this study yielded results that strongly suggest further metagenomic analysis should be conducted, using both longitudinal and seasonal study designs to provide statistically significant data to inform public health efforts.

The Crow Environmental Steering Committee has endorsed continued study, with a focus on both the Crow Fair swim hole site of the present study and upstream sites to determine the extent and potential sources of microbial contamination. The staff of the Crow Water Quality Project continue to educate the community on water quality and environmental health issues. Since the local tribal college is a two-year institution with limited facilities and resources, our use of the portable and relatively affordable MinION sequencing platform may serve as a proof of concept for introducing students at smaller tribal colleges to DNA sequencing technology as a means of monitoring water quality. Use of the MinION system may be applicable to the study of genomics in a teaching and research setting where the cost of other more expensive sequencing technologies is prohibitive.

## 5. Conclusions

Waterborne disease continues to threaten human health worldwide. Many regulatory agencies employ coliform testing of water as an indication of the extent of fecal contamination and disease risk. Even when the concentration of coliform bacteria is within an acceptable level, this method does not identify specific microbes that may be pathogenic at a very low dose-of-infectivity. In this study, we test the feasibility of using a highly portable DNA sequencing device, that may in the future be readily deployed for routine monitoring of water quality outside of research laboratory settings, for detection and metagenomic analysis of waterborne disease pathogens present in a river affected by fecal contamination from cattle ranching and leaking sewage systems. We demonstrate that even at an “acceptable” level of fecal coliform bacteria deemed to be safe for human recreational use of a river, seemingly rare and unexpected (for rural Montana) pathogens, such as *E. coli* O104:H4 and *V. cholerae*, as well as pathogens with a low dose-of-infectivity on the order of 1-10 cells, e.g., *E. coli* O157:H7, can be detected using metagenomic analysis.

As portable DNA sequencing devices continue to be refined and made more affordable, and as metagenomics software and analysis are fully integrated with these sequencing platforms, it can be envisioned that real time surveillance for water borne pathogens, virulence genes, and AMR gene markers will be incorporated into environmental monitoring to protect human health. The present study serves as a proof of concept of the utility of such an approach, by demonstrating the ability to detect not only pathogenic microorganisms, but also virulence and AMR genes. Use of traditional methods to screen for pathogens and phenotypic traits requires a targeted approach to test for specific agents and genes, and may require weeks or months to complete. Integrated DNA sequencing and metagenomic analysis, on the other hand, can be performed in real time, requiring only hours or days to complete an assessment for waterborne pathogens.

## Figures and Tables

**Figure 1 ijerph-16-01097-f001:**
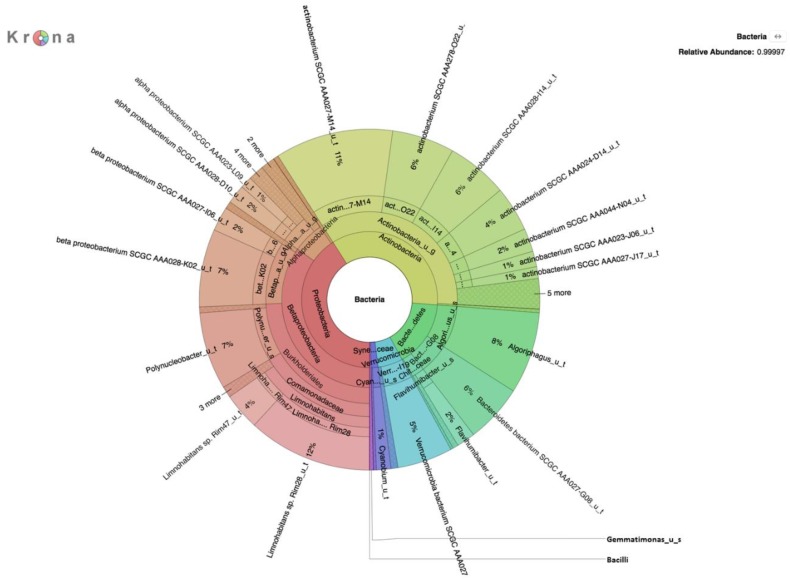
Krona plot of bacteria identified in the Little Bighorn River metagenome (DNA prepared without selection).

**Figure 2 ijerph-16-01097-f002:**
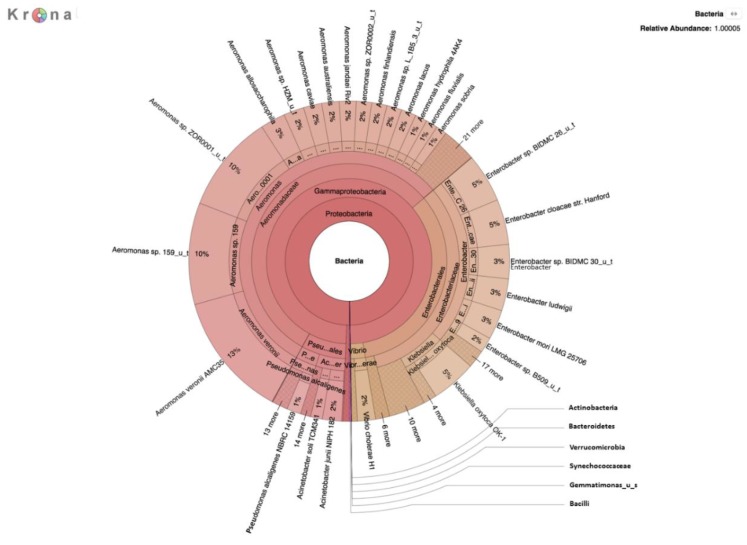
Krona plot of bacteria identified by metagenomic analysis of DNA prepared after selective growth on m-ColiBlue24 medium.

**Figure 3 ijerph-16-01097-f003:**
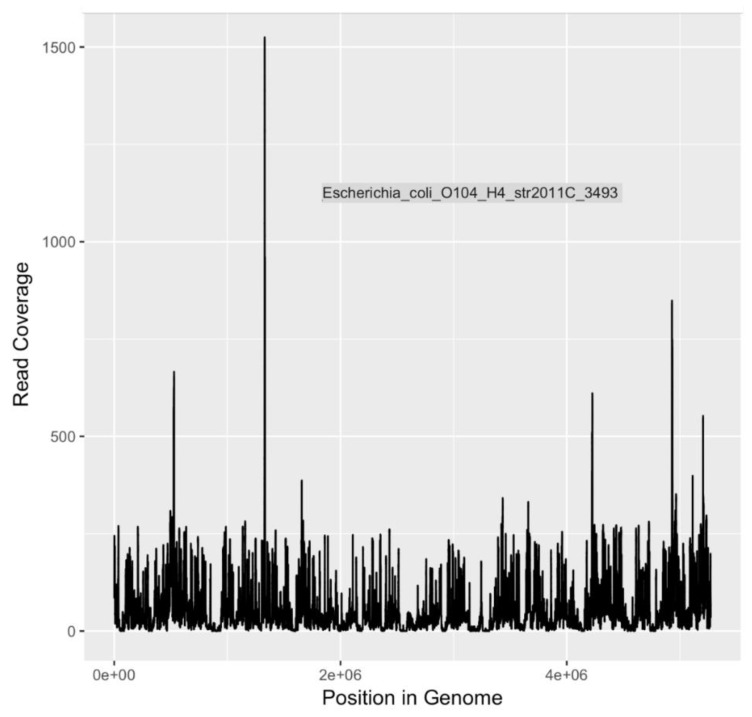
Sequencing coverage plot for *E. coli* O104:H4.

**Figure 4 ijerph-16-01097-f004:**
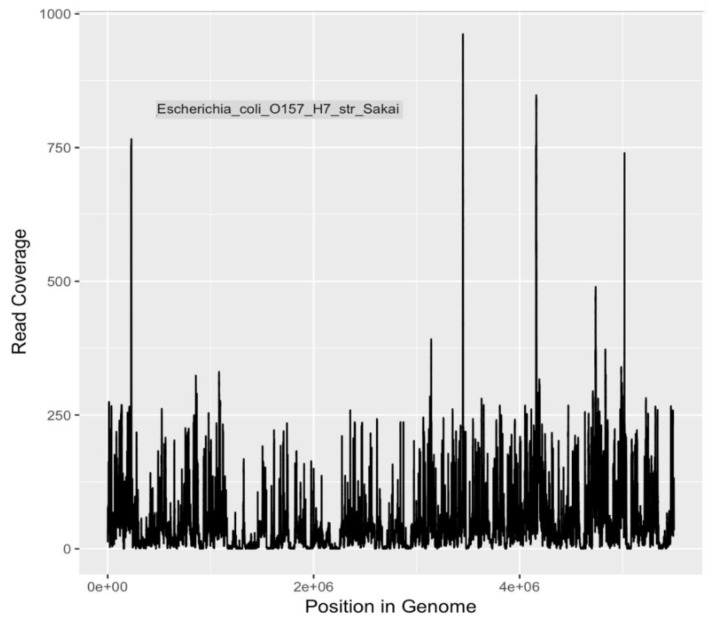
Sequencing coverage plot for *E. coli* O157:H7.

**Figure 5 ijerph-16-01097-f005:**
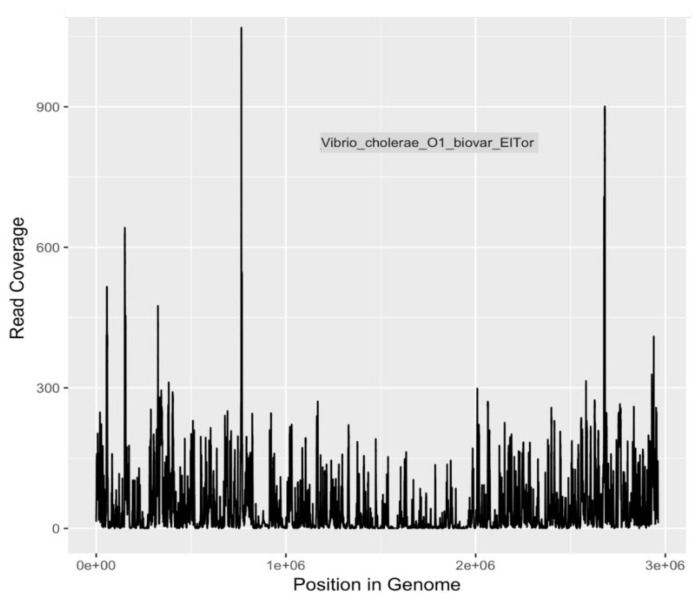
Sequencing coverage plot for *V. cholerae* O1 El Tor.

**Table 1 ijerph-16-01097-t001:** List of eukaryotic microbial genus and species, some of which are pathogenic, identified in the Little Bighorn River metagenome (DNA prepared without selection). WBD (waterborne disease) organisms have a known association with human disease.

Eukaryotic Genus	Eukaryotic Species	Number of Reads	Disease Association/WBD Organisms
*Acanthamoeba*	*Acanthamoeba polyphaga; Acanthamoeba palestinensis; Acanthamoeba quina; Acanthamoeba castellanii; Acanthamoeba healyi*	55	Infections of eye, skin, and central nervous system [34]
*Dictyostelium*	*Dictyostelium fasciculatum*	39	
*Guillardia*	*Guillardia theta*	6	
*Leishmania*	*Leishmania major; Leishmania donovani; Leishmania arabica; Leishmania infantum; Leishmania turanica; Leishmania aethiopica*	78	Infections of skin and internal organs [35]
*Oxytricha*	*Oxytricha trifallax*	109	
*Physarum*	*Physarum polycephalum*	24	
*Salpingoeca*	*Salpingoeca rosetta*	23	
*Symbiodinium*	*Symbiodinium minutum*	11	

**Table 2 ijerph-16-01097-t002:** List of fungal genera and species, some of which are known pathogens, identified as present in the Little Bighorn River metagenome (DNA prepared without selection).

Fungal Genera	Fungal Species	Number of Reads	Disease Association/WBD Organisms
*Amauroascus*	*Amauroascus niger*	292	
*Candida*	*Candida albicans; Candida dubliniensis*	37	Infections of the digestive system and vagina; invasive candidiasis [36]
*Chrysosporium*	*Chrysosporium queenslandicum*	198	
*Drechmeria*	*Drechmeria coniospora*	5	
*Magnaporthe*	*Magnaporthe oryzae*	5	
*Melampsora*	*Melampsora pinitorqua*	40	
*Orpinomyces*	*Orpinomyces sp*	61	
*Pleosporales*	*Pleosporales sp*	4	
*Rhizomucor*	*Rhizomucor variabilis*	10	Opportunistic infections [37]
*Rhizophagus*	*Rhizophagus irregularis*	108	
*Saccharomyces*	*Saccharomyces cerevisiae*	1	
*Trichoderma*	*Trichoderma longibrachiatum*	3	
*Ustilaginoidea*	*Ustilaginoidea virens*	1	
*Verticillium*	*Verticillium alfalfae*	1	

**Table 3 ijerph-16-01097-t003:** Bacterial community identified in the Little Bighorn River metagenome (DNA prepared without selection).

Bacterial Strain	Number of Reads	Disease Association
Acidovorax_sp_JHL_3	113	Sepsis [38]; catheter-associated bloodstream infection [39]
actinobacterium_SCGC_AAA023_J06	197	
actinobacterium_SCGC_AAA024_D14	503	
actinobacterium_SCGC_AAA027_M14	672	
actinobacterium_SCGC_AAA028_I14	580	
actinobacterium_SCGC_AAA044_N04	590	
actinobacterium_SCGC_AAA278_O22	811	
Aeromonas_salmonicida_subsp_salmonicida_A449	3	Fish pathogen [40]; isolated from human blood [41]; endophthalmitis [42]
Bacteroidetes_bacterium_SCGC_AAA027_G08	306	
beta_proteobacterium_CB	105	
beta_proteobacterium_SCGC_AAA027_I06	295	
beta_proteobacterium_SCGC_AAA027_K21	13	
beta_proteobacterium_SCGC_AAA028_K02	52	
Curvibacter_lanceolatus_ATCC_14669	173	
Exiguobacterium_acetylicum_DSM_20416	2	
Jonesia_denitrificans_DSM_20603	2	
Limnohabitans_sp_Rim28	2616	
Verrucomicrobia_bacterium_SCGC_AAA027_I19	141	

**Table 4 ijerph-16-01097-t004:** Bacterial species identified by metagenomic analysis of DNA prepared after selective growth on m-ColiBlue24 medium.

Bacterial Strain	Number of Reads	Disease Association/WBD Organisms
*Acinetobacter soli NIPH 2899*	1431	Bacteremia [44]
*Acinetobacter junii CIP 64 5*	1430	Septicemia [45]
*Aeromonas veronii AMC34*	82,139	Diarrhea [46]
*Aeromonas allosaccharophila strain CECT 4199*	60,447	
*Aeromonas sp 4287D*	46,956	
*Aeromonas australiensis strain CECT 8023*	46,643	
*Aeromonas fluvialis strain LMG 24681*	30,325	
*Aeromonas sobria strain CECT 4245*	22,042	
*Aeromonas sp AE122*	20,496	
*Aeromonas jandaei Riv2*	19,893	
*Aeromonas hydrophila subsp hydrophila ATCC 7966*	6444	
*Aeromonas caviae strain FDA MicroDB 78*	5495	
*Citrobacter braakii strain GTA CB04*	1010	Bacteremia [47]
*Cronobacter dublinensis subsp dublinensis LMG 23823*	1389	Opportunistic neonatal infection [48]
*Enterobacter ludwigii strain EN 119*	28,696	
*Enterobacter mori LMG 25706*	7371	
*Enterobacter asburiae L1*	3698	Opportunistic wound infection [49]
*Escherichia coli O104:H4 str 2011C 3493*	4390	Diarrhea, hemolytic uremic syndrome [50]
*Escherichia coli str K 12 substr MG1655 strain K 12*	2895	
*Escherichia coli O157:H7 str Sakai*	1250	Diarrhea, hemolytic uremic syndrome [51]
*Escherichia coli UMN026*	963	Urinary tract infection [52]
*Klebsiella sp BRL6 2*	6060	Nosocomial infections, hemorrhagic colitis, pneumonia, urinary tract infections [53,54]
*Klebsiella oxytoca HKOPL1*	1982	
*Klebsiella pneumoniae strain FDA MicroDB 64*	928	
*Leclercia adecarboxylata ATCC 23216 NBRC 102595*	1166	Bacteremia, soft tissue infection [55]
*Pseudomonas alcaligenes NBRC 14159*	3819	Opportunistic infections [56]
*Shigella flexneri 2a str 301*	1397	Shigellosis (diarrhea) [57]
*Shigella sonnei*	1137	
*Shigella boydii Sb227*	1124	
*Shigella boydii 5216 82*	863	
*Vibrio cholerae O1 biovar El Tor str N16961*	1388	Cholera [58]
*Vibrio albensis VL426*	1161	

**Table 5 ijerph-16-01097-t005:** Bacteriophages detected in the metagenomic analysis of DNA from filtered water sample after growth on selective m-ColiBlue24 medium.

Bacteriophage	Number of Reads	Gene Function and Disease Association
Aeromonas_phage_phiO18P	54	
Enterobacteria_phage_cdtI	39	
Enterobacteria_phage_Fels_2	8	
Enterobacteria_phage_fiAA91_ss	29	
Enterobacteria_phage_HK629	49	
Enterobacteria_phage_HK630	4	
Enterobacteria_phage_HK633	2	
Enterobacteria_phage_lambda	8	
Enterobacteria_phage_mEp043_c_1	7	
Enterobacteria_phage_mEp213	64	
Enterobacteria_phage_mEp460	33	
Enterobacteria_phage_P1	57	
Enterobacteria_phage_P2	20	
Enterobacteria_phage_P88	38	
Enterobacteria_phage_phiV10	9	
Enterobacteria_phage_YYZ_2008	18	
Salmonella_phage_RE_2010	18	
Salmonella_phage_SSU5	15	
Shigella_phage_SfII	20	O-antigen modification, enhancing antigen variation and resistance to host defense [59]
Shigella_phage_SfIV	37	O-antigen modification, enhancing antigen variation and resistance to host defense [60]
Stx2_converting_phage_1717	21	Encodes Shiga toxin, a virulence factor inhibiting protein synthesis in infected cells; cytotoxic [61]
Yersinia_phage_L_413C	22	

**Table 6 ijerph-16-01097-t006:** Antibiotic resistance markers identified by CosmosID analysis of DNA prepared from filtered water sample after selective growth on m-ColiBlue24 medium.

Resistance	Gene Name	Number of Reads
Ampicillin	*ampS*	26
Antibiotic Efflux	*mexB*	43
	*acrD*	37
	*acrB*	30
	*mdtF*	22
	*acrF*	18
	*mdtC*	13
	*mdtG*	10
Beta-lactamase	*blaCEPH-A3*	25
	*cphA4*	20
	*pbp2*	15
	*cphA1*	13
	*blaOXA-12*	12
	*cphA7*	10
Fluoroquinolone	*oqxB*	19
	*emrR*	10
Polymyxin Resistance	*pmrC*	16
	*arnA*	12

**Table 7 ijerph-16-01097-t007:** Virulence genes identified by analysis of the metagenome derived from DNA prepared after selective growth on m-ColiBlue24 medium.

Virulence Gene	Number of Reads	Gene Function and Disease Association
*E. coli* GENE f17G	108	Fimbrial adhesin; diarrhea [62]
*E. coli* GENE gad	627	Homeostasis and acid resistance [63]
*E. coli* GENE iss	78	Increased serum survival; extraintestinal infection [64]
*Pasteurella multocida* GENE tetH	80	Tetracycline resistance; bacteria cause a variety of diseases in mammals and birds, and opportunistic infections in humans [65,66]
*Pasteurella multocida* GENE tetR	29	
*Vibrio cholerae* GENE vasH	83	Type VI secretion system; promotes competitive advantage by killing other cell types, and fosters horizontal gene transfer to enhance the evolution of virulence and antibiotic resistance [67]
*Vibrio cholerae* GENE vgrG-3	111	
*Vibrio cholerae* Gene VCA0107	19	
*Vibrio cholerae* Gene VCA0109	48	
*Vibrio cholerae* Gene VCA111	51	
*Vibrio cholerae* Gene VCA0121	121	
*Yersinia pestis* GENE ybtE	59	Yersiniabactin iron acquisition system, to obtain iron from the host during infection [68]
*Yersinia pestis* GENE ybtQ	132	
*Yersinia pestis* GENE ybtX	101	

**Table 8 ijerph-16-01097-t008:** Sequencing coverage calculations for *E. coli* O104:H4, *E. coli* O157:H7, and *V. cholerae* O1 biovar Eltor.

Bacterial Species	Number of Bases Covered	Reference Genome Length	% Coverage along Reference Genome	Depth of Coverage (mean)	Depth of Coverage (st dev)
*E. coli* O104:H4	5,226,510	5,437,407	96.3	52.2	61.8
*E. coli* O157:H7	5,332,768	5,594,477	95.3	50.2	60.7
*V. cholerae* O1 biovar ElTor	3,770,896	4,033,464	93.5	35.8	58.3

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
