# Peer review of "Metagenomic Profiling of Microbial Pathogens in the Little Bighorn River, Montana"

_ijerph, 2019, doi:10.3390/ijerph16071097_

Round 1

Reviewer 1 Report

The authors detailed a very well thought out experimental design with results that have the potential to significantly impact the study of environmental exposures. Since waterborne pathogen exposures are a major global public health issue, studies of this type will help identify etiological agents of waterborne pathogen infections while producing new tools for public health officials to combat these exposures and subsequent infections.

The manuscript is well researched and written. However, problems with the numbering of the references should be corrected. Beginning with reference # 8 which does not exist. One example is Line 60: Hamner is not reference # 17 but # 18. Start here on Line 60 and correct the other references, particularly those in the tables to ensure that the match.

Line 272: correct enteraggregative to "enteroaggregative"

Author Response

20 March, 2019

To: IJERPH Editorial Office

From: Steve Hamner and Timothy E. Ford

RE: cover letter describing responses to all referees’ comments

We sincerely thank both reviewers for their informative comments and suggestions for improving the manuscripts. We very much appreciate the time and effort spent.

Changes to the manuscript have been highlighted in yellow.

Note: All references to line numbering reflect the original document; line numbering has been altered due to editing and addition of text.

Response to Reviewer 1:

The manuscript is well researched and written. However, problems with the numbering of the references should be corrected. Beginning with reference # 8 which does not exist. One example is Line 60: Hamner is not reference # 17 but # 18. Start here on Line 60 and correct the other references, particularly those in the tables to ensure that the match.

The formatting errors for references, several of which appear to have been introduced during reformatting of the manuscript by the publisher’s office, have been corrected.

Line 272: correct enteraggregative to "enteroaggregative"     corrected

Response to Reviewer 2:

General comments

The authors investigated the microbial quality of the Little Bighorn River in Montana by performing a metagenomic approach. The paper is well written but needs to address certain concerns before it can be accepted for publication in IJERPH. For example, the use of a 0.45 uM membrane for the total genomic isolation is also a problem. Although it is generally believed that these filters can retain most bacteria, about 19 different taxa have been shown to pass even through 0.2 micron filters (See Broad diversity of viable bacteria in 'sterile' (0.2 microm) filtered water)). Thus, I think that the authors might have underestimated the pathogenic microbial load in their analysis. I acknowledge that the authors have explained their choice of membrane at Lines 261 – 264, however, I think this was for the selective culture and not for the whole river water sample. I think concentration through centrifugation for example could have yielded better results.

We have made a number of revisions in the ms to address this Reviewer’s concern, and have elaborated on a number of issues that come into play when regarding heavily polluted environmental water samples such as those with which we routinely work. First, our work follows the guidelines for current EPA-approved water testing protocols (including use of 0.45 micron filters for the m-ColiBlue24 method); thus, we made a carefully considered decision to use 0.45 micron filters in the present study. Additionally, the particulate matter (dirt and debris) present in our water samples caused complete clogging of 0.2 micron filters, even when well under 100 mls of water was filtered. Accordingly, use of multiple filters, the number of which often is determined by trial and error, would be required to trap enough cells to generate enough DNA for analysis. Simply obtaining DNA with “good” A260/A280 spectophotometric ratios may not be adequate. In our experience, the presence of inhibitors of unknown identity after filtration of environmental water samples frequently interfere with PCR reactions and DNA sequencing. Producing a sufficient quantity of high quality DNA without inhibitor activity can require extensive cleanup above and beyond the use of commercially available DNA extraction kits and published protocols and that additional cleanup entails substantial loss of DNA. The instruction manual for the PowerWater kit we used discusses these challenges at length, and recommends use of 0.45 micron filters for processing of water with significant turbidity.

We acknowledge that some bacterial members of the consortium may have been “missed” through use of the larger pore sized filters (0.45 micron). More importantly, in showing in the present study the success of employing metagenomic analysis to identify a large number of pathogens, even with the use of 0.45 micron filters, we readily acknowledge that with future study and refinement of methods, 0.22 micron filters could eventually be used to identify an even larger range of pathogens not detected in the present study.

We have added the following technical note statement near the beginning of M&M to lend understanding to some of these issues: “(Technical notes: The PowerWater kit was chosen in part due to its incorporation of reagents to remove inhibitors that may interfere with downstream PCR and DNA sequencing reactions. Choice of 0.45 micron filters as opposed to 0.22 micron filters was due to turbidity of the water samples.  We followed the manufacturer’s recommendation to use 0.45 micron filters for turbid water samples to reduce clogging and allow a greater volume of water to be filtered than would be possible with the more restrictive 0.22 micron filters. We readily acknowledge that this choice may have reduced the variety of bacteria detected, since during the initial stages of filtering, smaller bacteria would be lost in the larger 0.45 micron pores but also are aware that as the filters clogged, many smaller cells should have been captured.).”

AbstractLine 31: Please change “anti-microbial” to “antimicrobial”

corrected

Lines 32 – 34: Metagenomics is not “new” in the determination of microbial contamination as many studies have reported on it already. This makes the conclusion weak. It would be better to conclude by giving the implications of the study findings to the health of the population (look at the introductory statement in the abstract) and recommend that metagenomics be included as a routine tool for in-depth understanding of the microbial contamination of waterways.

Sentence in lines 32-34 is reworded to reflect the reviewer’s recommendation. Sentence now reads: “It is concluded that metagenomics provides a useful and potentially routine tool for identifying in an in-depth manner microbial contamination of waterways and, thereby, protecting public health.”

Materials and methods

It would be good to have a map of the sampling area indicating the sampling point(s). What is the approximate population of the study area?

We have added a statement of latitude/longitude coordinates, and population of Crow Agency has been added. Since only a single sampling site was studied in the present study, we believe that inclusion of a map is non-essential for the present publication. We also note that a detailed map is not readily available due to sovereignty and privacy issues on the reservation and the need to always be cognizant of Tribal wishes. Obtaining approval from the Crow Environmental Health Steering Committee would require our requesting permission from the Committee members who only meet once a month or so to consider such a request and would delay our resubmission of the manuscript. In future source tracking study of the contamination identified, inclusion of a map would certainly be useful. The statement we have added to the first paragraph of the Material and Methods section is, “The swim hole is located at latitude/longitude of 45o36’1”N. 107o27’12”W, and is a popular summer recreational site used by children and adults of the population of ca. 1,600 residents of Crow Agency.”

Line 101: 0.45? Something is lacking.       

Corrected to add the unit of measurement (μm)

Line 110 – 120: Why was the selective isolation of EHEC carried out if at the end, the entire membrane, rather than selected presumptive colonies were to be chosen for the metagenomic analysis (Line 117 – 120).

We selectively re-grew an m-ColiBlue24 filter’s worth of cells through replica plating on CHROMagarO157 agar, simply to check for appearance of mauve-colored colony growth as a test for putative E. coli O157 bacteria, but did not then proceed to actually isolate those bacteria. Because the two growth media differ in chemical composition and are used for completely different purposes, if we had conducted metagenomic analysis of growth from the CHROMagarO157 medium, we may well have failed to identify many of the bacterial gene signatures found through analysis of the growth resulting from culture on the m-ColiBlue24 media.

Line 131: the eaeA gene is not always indicative of the EHEC, especially if identified alone. The authors may want to take this into consideration.

To clarify our testing strategy here, we added the sentence, “Presence of eae is characteristic of both EHEC and EPEC; on the other hand, EHEC contains Stx genes while EPEC lacks Stx genes.”

Results

Line 144 (not 114): I recommend the authors verify this claim. Some case reports of Aeromonas salmonicida in humans do exist in literature. Also, other members of the Aeromonas genes are pathogenic to humans. Similarly, Acidovorax spp have been involved in hospital-acquired infections. So, it may not be correct to say that “no bacterial genera”

Our original literature search failed to reveal references on Aeromonas salmonicida (subspecies salmonicida A449) in human disease, and a specific search for the Acidovorax_sp_JHL_3 also failed to reveal references to human disease. To address the reviewer’s note, we conducted a new search removing the subspecies designations, and now include 2 references for each of Aeromonas salmonicida and Acidovorax,. Wording has been revised: “Bacteria of concern to human health, Acidovorax and Aeromonas salmonicida, were also identified (Table 3).”

Line 178: Why did the authors not check for ARG and VG markers in the non- selective native river water metagenome?

Identification of gene markers is based on the database and algorithms used by CosmosID; these markers were simply not identified by the analysis.

Line 181 – 184: “Two types of Shigella....hemolytic uremic syndrome.” Should be moved to the discussion.

As suggested by this Reviewer, the sentence has been moved to the Discussion and an introductory sentence added; text now reads: “Of relevance to human health, bacteriophage-encoded genes that enhance the pathogenicity of host bacteria were also detected. Two types of Shigella-specific phage, SfII and SfIV, allow for O antigen modification and increased antigen variation [55, 56]. The Stx2 converting phage of E. coli O157:H7 and other related Shiga toxigenic E. coli (STEC) encodes the Stx2 protein, an important virulence factor causing lysis of host cells and contributing to hemolytic uremic syndrome [57].”

Line 194 and line 195: ...were both undetected......were both detected.....      

corrected

Discussion

Lines 228 – 240: The authors rightly say that metagenomics allow for the detection of organisms that would not otherwise be detected through conventional culture methods. However, in many instances in the results section, they reported that bacterial pathogens, for example, were only detected in the selective culture method. What could have accounted for this difference as one would expect more from the whole samples than from the culture samples.

Please see lines 264-266 (new line numbering): Lack of detection of E. coli by metagenomic analysis without selective growth is attributed to the overwhelming abundance and diversity of non-E. coli microorganisms that were present.

Lines 246 – 260: Are the authors saying that the selective enrichment is more sensitive than the metagenomics approach and that is why E. coli could only be detected through selective culture? This may be contradictory to existing literature, and as mentioned in the introduction of this manuscript, on the superior sensitivity of molecular methods over culture-based methods for bacterial identification.

Please see the sentence in lines 264-266 noted in the previous comment, as well as the next sentence in lines 266-269 (“The proportion of E. coli present in the water samples was representatively small in comparison to the high microbial load on selective media, evidenced by results of the analysis of the m-ColiBlue24-derived metagenome, revealing Gammaproteobacteria and coliform bacteria in significant abundance.”); metagenomics is more sensitive; it is only through/after selective enrichment that there are now enough E. coli and related pathogenic bacterial gene sequences for the metagenomics analysis to detect these sequences. Our approach makes use of a combination of both metagenomics and enrichment to allow for identification of the E. coli and related pathogens’ gene signatures. This would not have been possible using either the selective culture or metagenomics approach alone.

Lines 296 – 301: This section contains information that appears to be contractor. Genes related to O157 were detected. These bacteria were isolated on “selective O157” media. Yet Line 298 says “Although these bacteria were not directly isolated by culture....”. The section may need some rephrasing.

We have rephrased a sentence as follows to clarify (new line numbering 313-316): Presence of Shiga toxin gene markers indicated by both PCR (data not shown) and metagenomic analysis, as well as mauve-colored colony growth on ChromagarO157 medium, a differential/selective medium and indicative test for O157:H7, provide both genetic and phenotypic evidence for continued presence of E. coli O157:H7 bacteria in the Little Bighorn River.

Conclusion

This section sounds as though the aim of the work was to we test the feasibility of using the Oxford Nanopore MinION DNA sequencing device.....If this is the case, then the topic should be modified.

The sentence in question has been reworded to reflect the theme of using a highly portable DNA sequencing device (not confined to just use of the MinION device since other portable systems may become available in the future), that with improvement may be employed for routine public health testing of water quality, to more comprehensively identify waterborne disease pathogens and threats. The newly reworded sentence (New line 361): “In this study, we test the feasibility of using a portable DNA sequencing device, that may in the future be readily deployed for routine monitoring of water quality monitoring outside of research laboratory settings, for detection and metagenomic analysis of waterborne disease pathogens present in a river affected by fecal contamination from cattle ranching and leaking sewage systems.”

Reviewer 2 Report

General comments

The authors investigated the microbial quality of the Little Bighorn River in Montana by performing a metagenomic approach. The paper is well written but needs to address certain concerns before it can be accepted for publication in IJERPH.

For example, the use of a 0.45 uM membrane for the total genomic isolation is also a problem. Although it is generally believed that these filters can retain most bacteria, about 19 different taxa have been shown to pass even through 0.2 microm filters (See Broad diversity of viable bacteria in 'sterile' (0.2 microm) filtered water)). Thus, I think that the authors might have underestimated the pathogenic microbial load in their analysis. I acknowledge that the authors have explained their choice of membrane at Lines 261 – 264, however, I think this was for the selective culture and not for the whole river water sample. I think concentration through centrifugation for example could have yielded better results.

Specific comments

Abstract

Line 31: Please change “anti-microbial” to “antimicrobial”

Lines 32 – 34: Metagenomics is not “new” in the determination of microbial contamination as many studies have reported on it already. This makes the conclusion weak. It would be better to conclude by giving the implications of the study findings to the health of the population (look at the introductory statement in the abstract) and recommend that metagenomics be included as a routine tool for in-depth understanding of the microbial contamination of waterways.

Introduction

This section is well-written and straight to the point.

Materials and methods

It would be good to have a map of the sampling area indicating the sampling point(s). What is the approximate population of the study area?

Line 101: 0.45? Something is lacking.

Line 110 – 120: Why was the selective isolation of EHEC carried out if at the end, the entire membrane, rather than selected presumptive colonies were to be chosen for the metagenomic analysis (Line 117 – 120).

Line 131: the eaeA gene is not always indicative of the EHEC, especially if identified alone. The authors may want to take this into consideration.

Results

Line 114: I recommend the authors verify this claim. Some case reports of Aeromonas salmonicida in humans do exist in literature. Also, other members of the Aeromonas genes are pathogenic to humans. Similarly, Acidovorax spp have been involved in hospital-acquired infections. So, it may not be correct to say that “no bacterial genera”

Line 178: Why did the authors not check for ARG and VG markers in the non-selective native river water metagenome?

Line 181 – 184: “Two types of Shigella….hemolytic uremic syndrome.” Should be moved to the discussion.

Line 194 and line 195: …were both undetected……were both detected…..

Discussion

Lines 228 – 240: The authors rightly say that metagenomics allow for the detection of organisms that would not otherwise be detected through conventional culture methods. However, in many instances in the results section, they reported that bacterial pathogens, for example, were only detected in the selective culture method. What could have accounted for this difference as one would expect more from the whole samples than from the culture samples.

Lines 246 – 260: Are the authors saying that the selective enrichment is more sensitive than the metagenomics approach and that is why E. coli could only be detected through selective culture? This may be contradictory to existing literature, and as mentioned in the introduction of this manuscript, on the superior sensitivity of molecular methods over culture-based methods for bacterial identification.

Lines 296 – 301: This section contains information that appears to be contractor. Genes related to O157 were detected. These bacteria were isolated on “selective O157” media. Yet Line 298 says “Although these bacteria were not directly isolated by culture….”. The section may need some rephrasing.

Conclusion

This section sounds as though the aim of the work was to we test the feasibility of using the Oxford Nanopore MinION DNA sequencing device…..If this is the case, then the topic should be modified.

Author Response

(The authors gave the same response as above.)

Round 2

Reviewer 2 Report

The authors have attended to all the comments raised. It would be interesting to carryout further works in that catchment. QMRA studies could be very valuable in that regards.